# Organising polypharmacy: unpacking medicines, unpacking meanings—an ethnographic study

Deborah Swinglehurst , Nina Fudge

Wolfson Institute of Population Health, Barts and The London School of Medicine and Dentistry, Queen Mary University of London, London, UK

**Correspondence to**
Professor Deborah Swinglehurst;
d.swinglehurst@qmul.ac.uk

## ABSTRACT

**Objectives** We explore how older patients affected by polypharmacy manage the 'hidden work' of organising their medicines, how they make sense of this work and integrate it into their lives.

**Design and setting** Ethnographic study observing patients over 18–24 months in patients' homes, general practice and community pharmacy, in England, UK.

**Participants and methods** Ethnographic case study including longitudinal follow-up of 24 patients aged 65 or older and prescribed ten or more items of medication. Our dataset includes: 562 hours of ethnographic observation across patients homes, community pharmacies and general practices; 47 audio-recorded interviews with patients about their lives and medicines practices; cultural probes (photographs, body maps, diaries and imagined 'wishful thinking' conversations); fieldnotes from regular home visits; telephone calls, and observation/video-recording of healthcare encounters. We apply a 'practice theory' lens to our analysis, illuminating what is being accomplished, why and by whom.

**Results** All patients had developed strategies and routines for organising medicines into their lives, negotiating medicine taking to enable acceptable adherence and make their medicines manageable. Strategies adopted by patients often involved the use of 'do-it-yourself' dosette boxes. This required careful 'organising' work similar to that done by pharmacy staff preparing multicompartment compliance aids (MCCAs). Patients incorporated a range of approaches to manage supplies and flex their regimens to align with personal values and priorities. Practices of organising medicines are effortful, creative and often highly collaborative. Patients strive for adherence, but their organisational efforts privilege 'living with medicines' over taking medicines strictly 'as prescribed'.

**Conclusions** Polypharmacy demands careful organising. The burden of organising polypharmacy always falls somewhere, whether undertaken by pharmacists as they prepare MCCAs or by patients at home. Greater appreciation among prescribers of the nature and complexity of this work may provide a useful point of departure for tackling the key issue that sustains it: polypharmacy.

## Strengths and limitations of this study

► Gathers rich qualitative data from a range of community settings and is grounded in the everyday experiences of participants.

► Reveals decision-making and practices around medication management which are often hidden from view and unknown to professionals who prescribe and dispense medicines.

► Uses innovative qualitative methods to explore aspects of everyday life which might be difficult to reach using more conventional approaches.

► Prioritises depth of analysis over breadth, enabling richness of conceptual insights but limiting generalisability.

► Patients' willingness to take part in our research might reflect their commitment to their medicines.

## INTRODUCTION

Polypharmacy is one of three key action areas in the WHO Global Challenge 'Medication without Harm'.[1] The prevalence of polypharmacy is increasing. In one Scottish study, 20% of adults were dispensed 5 or more drugs and 6% of adults were dispensed 10 or more drugs.[2] Patients receiving 10 or more drugs are at particular risk of harm, so-called 'high risk' polypharmacy.[3] Guthrie's study showed a tripling in the prevalence of this phenomenon between 1995 and 2010.[2–5] Taking medication reliably and safely is demanding for patients, especially when complex medication regimens are involved.[6 7] Supporting patients in taking their medication is a key policy and professional commitment, enshrined within 'medicines optimisation': a 'person-centred approach to safe and effective medicines use, to ensure people obtain the best possible outcomes from their medicines'.[3 8 9] Medicines optimisation is situated within an objective 'rational use of drugs' paradigm, which emphasises judicious prescribing by well-informed professionals of well-studied drugs to well-informed patients for well-defined conditions.[10] Achieving medicines optimisation is assumed to hinge centrally on a process of 'shared decision making' within the clinical consultation. This is emphasised in the National Health Service England 2020/2021 Network Contract

Directed Enhanced Service (DES) specification which requires primary care networks (PCNs) to invite patients who are prescribed 10 or more medications for a structured medication review with an appropriately trained clinical pharmacist or general practitioner.[11]

Notwithstanding the importance of medication reviews, most decision-making about medicines taking, and almost all of patients' medication practices,[12] take place beyond the clinic, usually in patients' homes. In the clinic, medicines are typically conceptualised as biomedical technologies (a disease, or risk of disease, is identified and a medicine is prescribed to ameliorate symptoms or reduce risk). But medicines can also be conceptualised as socially embedded phenomena, as representations that carry meanings and shape social relations.[13] Medicines are material *things* of therapy which have a social life,[14] and this becomes especially salient when medicines enter patients' homes and daily lives. Here, patients construct a 'lay pharmacology' of personally relevant notions of safety, side effects and efficacy which both converge with, and diverge from, conventional biomedical perspectives.[15] Medication practices become sites of moral evaluations of self, society and pharmaceuticals; peoples' relationship with medicines offers insight into their identities, and into these moral stories of self and society.[16] Consumption of medicines becomes a matter of embodied, purposeful, situated accomplishment,[7 17] involving a particular kind of *work* or expertise as patients devise ways to incorporate their medicines into their daily lives. Patients have to 'do' medicines adherence.[7]

Under this view, taking medicines is not only a matter of cognition and decision-making (shared or otherwise). It is also a very practical matter of *organising*, of managing medicines *into* daily life and establishing routines. Indeed, one of the key motivations for establishing routines is to *minimise* the cognitive demands involved in carrying them through. A balance must be struck. As medicines routines become more reliable (and the attention invested in them reduces), the unintentional risk of missing doses increases, so patients must devise further strategies to anticipate this possibility.[18 19] Huyard *et al* studied unintentional non-adherence among patients prescribed single items of long term medication and found that patients used a range of strategies to support their adherence.[19] These included *anchoring* (scheduling pill-taking with another pre-existing routine), *spatial positioning* of pills in ways that make pill-taking more reliable, and methods for *verifying* that pill taking has occurred.[19]

When polypharmacy involves large numbers of drugs, it presents particular challenges to both professionals and patients. If a genuinely *person-centred* approach to medicines optimisation is sought, research is needed which illuminates these challenges, by attending to professionals' and patients' subjective, context-specific evaluations of medicines, their lived experiences and everyday practices. When patients devise routines to support their medicines taking, they are invested in biographical work, in what Huyard describes as the 'microintegration of their

illness deep into patients' everyday lives'.[19] Our research adopts an ethnographic approach to explore these practices. Ethnography is well suited to investigating complex multifaceted phenomena by making tacit knowledge and practices visible.[20] It involves immersion in the field, an attitude of 'appreciation' towards the social world *as it is*, presentation of 'first hand' accounts of observations, and acceptance that research is an active process in which accounts of the world are constructed through selective observation and theoretical interpretation.[21]

In this paper, we focus primarily on the practices of older adults who live at home and are prescribed 10 or more separate items of medication. To set the scene, we begin in a community pharmacy, with a brief ethnographic description of practices we observed as professionals prepared multicompartment compliance aids (MCCAs) or 'dosette' boxes for some patients affected by polypharmacy. This draws attention to the organisation of polypharmacy as a form of 'work'. We then shift our ethnographic gaze from the professional context into the private context of patients' homes to address the question: 'How do patients organise their medicines and medicine-taking?' We explore the meanings embedded in these everyday acts of organising and illuminate who is doing the work and what this work involves. We show how patients draw on personal, social, material and hermeneutical resources in their efforts to adhere to their medicines within the context of their daily lives, and how their lives are shaped by, and in turn shape, their medicines regimens.

## METHODS

The study reported here is part of the wider APOLLO-MM study (Addressing the Polypharmacy Challenge in Older People with Multimorbidity). The methods have been described in detail elsewhere.[22] Table 1 summarises our data collection across general practices, community pharmacies and patients' homes. Our approach has enabled us to build a rich picture of our patient participants' experiences in both personal and professional contexts. We have used O'Brien et al's Standards for Reporting Qualitative Research (SRQR).[23]

### Setting and data collection

We recruited a purposive sample of 24 patients, 14 from two inner city general practices in an area of high deprivation, 10 from a suburban general practice in a relatively prosperous city. Patients were aged 65 or over, living at home and were prescribed 10 or more regular items of medication which were dispensed by our study pharmacies. The sample reflects diversity of age (65–94); gender (11 men, 13 women); socioeconomic status; number of items of regular medication; comorbidities. We excluded patients unable to speak adequate English. All patients gave written informed consent to participate in the study. Our sampling was guided by Malterud's notion of information power and reflects our detailed data collection

| Table 1 | Summary of data collection methods |
|---|---|
| Method/approach | Details of data collection |
| Organisational ethnography of GP practices | 184 hours observation in three GP practices, observing professional practices relevant to prescribing including shadowing staff and 6 interviews |
| Organisational ethnography of community pharmacies | 140 hours in four community pharmacies, observing professional practices relevant to polypharmacy, including shadowing staff and 21 interviews[29] |
| Longitudinal follow-up of 24 patients | Ethnographic observation in patients' homes (66 home visits) and regular follow-up conversations by telephone (258 telephone calls). We accompanied participants to selected hospital and GP consultations and on trips to the local pharmacy (29 clinical consultations of which 18 were video recorded). |
| Biographical narrative interviews with patients | 24 in-depth narrative interviews using Wengraf's biographic-narrative-interpretive method,[42] eliciting a narrative with one opening question ("please tell me the story of your life since you were first advised to take medicines") and using the participant's account to elicit further narratives. |
| In-depth interviews with patients about their medicines | 23 in-depth interviews focused on participants' medicines and medicines practices (one participant withdrew from study due to a decline in health before this interview could be completed). |
| Cultural probes with 14 of the patient participants. | Cultural probes encouraged participants to depict their lives with their medicines through a number of optional activities[43]: <br> ▶ giving participants a camera and asking them to take pictures of what their medicines mean to them <br> ▶ drawing body maps of how they are feeling and how their medicines make them feel <br> ▶ completing diaries of social contact <br> ▶ 'wishful thinking': participants imagined conversations they would like to have with a healthcare professional about their medicines |

methods and multiple opportunities to interact with study participants.[24] Data were collected between September 2017 and July 2020.

Each patient constituted a 'case'. We followed our participants' lives and experiences over an 18–24 month period of ethnographic observation involving a range of data collection activities (see table 1).

All interviews took place in participants' homes and were audio recorded. Two researchers (NF, a social anthropologist and DS, an academic GP, both experienced ethnographers in healthcare settings) undertook data collection and analysis (author initials in the text refer to which researcher conducted interviews or documented field notes). Two patients withdrew from the study after 8 and 11 months of follow-up.

### Analysis

Fieldnotes and interview transcripts were shared between the researchers and we maintained a shared digital reflexive journal using Evernote to document analytic memos and ongoing analytic insights, in addition to regular data analysis meetings. We used QSR NVivo V.12 qualitative data analysis software to manage the data,[25] and adopted an interpretive approach to analysis. Our analysis is informed by practice theory[26 27] which conceptualises practices as arrays of embodied human activity and draws attention to assemblages of people, technologies, artefacts and their interconnections in context, including shared practical understandings, 'know-how', skills, tacit understandings and dispositions. All names

reported in this paper are pseudonyms, and case narratives have been adapted to assure anonymity of research participants.

### Patient and public involvement

We have an online patient panel of five members and a project advisory group with 11 members: lay chair; academics; health professionals; representation from Age UK; two patient members. Patients were involved in: proposal development; designing participant materials/ project website (www.polypharmacy.org.uk); application for ethical approval; project launch event; piloting interviews; study design and conduct.

### STUDY FINDINGS

Medications are much more than material objects with physiological effects; they are also representations that carry meanings and shape social relations as they evolve in conjunction with individuals and collectivities.[10]

We begin by presenting an ethnographic account of the professional work of organising polypharmacy. This account focusses on the preparation of multicompartment compliance aids (MCCAs or 'dosettes'). These were dispensed to 4 of our 24 research participants at the outset of our study; 8 participants received them by the end of our ethnographic follow-up. MCCAs are typically dispensed to patients affected by polypharmacy who are having difficulty managing their medicines

---

**Box 1 Ethnographic account of multicompartment compliance aids (MCCA)/dosette preparation in Willow Pharmacy**

Willow Pharmacy is the 'dosette hub' for the Woodland Independent Pharmacy group. The work of preparing dosettes is called 'being on production' and is a technology-supported routine involving a full-time dosette robot and its human co-workers, typically three pharmacy technicians, also working full-time to ensure supply keeps up with demand. The robot cannot work unsupervised and needs constant attention. Demand for dosettes is increasing every month, as displayed on a handwritten poster on the wall in the 'production area' where a running tally is kept. The pharmacy technicians are engaged in a constant, well-coordinated round of stocktaking, 'de-blistering' drugs, 'replenishing' the robot and running each production session. Deblistering is the process by which the staff prepare the medicines for the robot, as it cannot process medicines from their original packs. Pills are 'popped' out, one by one, from their blister packaging into containers which slide into the robot. Some staff use a deblistering machine for this, pulling a large metal lever to press the pills out of their packets into interim containers for checking by the pharmacist prior to replenishment. Others do it manually. Technicians work at speed and use a wide range of technologies including: computer programmes; Excel spreadsheets, often printed out; written log books; printed cards; scribbled post-it notes. All contribute to maintaining the smooth flow of routines and a sense of predictability as they generate dosettes in 4-weekly batches ready to stack on the shelves awaiting final checking. Medicines are referred to as 'fast lines' (ie, there is a lot of demand for them by the pharmacy's clients) or 'slow lines' (ie, less demand). This categorisation is an important component of decision-making regarding whether a particular medicine warrants inclusion in the robot. 'Slow lines' may be designated 'externals' (meaning that the medicine is added to a dosette box by hand once the robot has done its job). The integrity of medicines may be at risk if they are left out of their packaging for too long. Expensive lines are also dealt with as 'externals' so that the pharmacy can remain agile in its response to market price fluctuations. Despite everyone's best efforts to rationalise production, there are inevitably plenty of exceptions and contingencies to deal with. We often see filled dosettes recovered again from the shelves as staff attend to unanticipated changes in patients' prescriptions. This entails a painstaking process of using tweezers as a tool to isolate, break off and remove the backing from relevant cells of the dosette, then carefully removing and/or adding the relevant medicines, sealing the cell up again and attending to all the associated paperwork and electronic documentation. If the technician in the 'dosette corner' receives too many requests for changes of this kind in quick succession, the job quickly becomes very stressful.

independently. In our observations of the 'backstage' areas of community pharmacies, we were struck by the scale and complexity of this work.[28] Box 1 presents a brief description based on one community pharmacy site (for a more detailed account of our observations in community pharmacy see Fudge and Swinglehurst 2021).[29]

This account offers some insight into the scale and complexity of the organisational work that polypharmacy presents and the range of professional considerations that are brought to bear as routines are developed to ensure that work proceeds as swiftly and safely as possible. It contextualises polypharmacy as the locus of *work*. In the

remainder of this paper, we consider the organisational work of our patient participants. We focus on patients who were dispensed all of their medications in their original packaging, where the burden of organisational work sits squarely with the patient. At the outset of our study, 20 of the 24 participants received all their medication in original packaging.

Many participants expressed a sense of obligation and resignation with regards to taking medicines 'as prescribed', often expressing medicine-taking as a non-negotiable aspect of their lives. At the same time, they readily adapted their regimens to make them manageable. Their organisational efforts reflect a balance of these two broad commitments.

I have to take them, that's is. I have to accept it…I mean if I was on less, yeah, I'd be happy, but I'm not, so I just have to take them and that's it (Biographical interview, Marco, DS)

I'm supposed to take that [carbocisteine] two tablets three times a day. I don't. Morning and evening. I usually forget the afternoon one, and it's possibly insufficient, but I have to balance one inconvenience with another inconvenience.

And later he goes on:

It [taking medicines] always takes place after breakfast, although breakfast can be anything between eight and ten. I know that I should be taking them [the 'morning' tablets] at a specific time each day in preference, but in life it doesn't always happen like that. (Medicines interview, Charles, DS)

Participants invested considerable time and effort into organising their medicines into their lives, motivated by their striving to adhere to their medicines regimens:

*I'm actually very methodical about it and I almost never miss any, which I believe is quite unusual, but it's because I've got this routine that I feel comfortable…it's been going on exactly the same for a number of years. (Medicines interview, Elaine, NF)*

*Everything's in the kitchen…when they're on top of the microwave, I can't forget them. (Medicines interview, Maria, NF)*

*I try to be organised, because it's easier to do that than getting stuff out every day. I'd find that very confusing. (Medicines interview, Marian, DS)*

*When I ask Charles if he will show me his medicines he gets up quickly from his chair and moves towards the kitchen. I don't feel able to follow him there. He reappears with a little tray with a lip round the edge about an inch high. In it there is a very neatly organised set of medicine boxes lined up, with some sachets of Laxido [a laxative] propped up along one side. It looks very orderly and I wondered how he managed to find a tray that is such a perfect fit for his medicines boxes, or indeed what would happen if he were to stop half of them as they wouldn't tessellate and hold their place quite as they do in this neat arrangement. There is one open strip of capsules adjacent to the Laxido sachet, but otherwise everything is kept in its original box. He tells me that they are kept in a kitchen cupboard and later when I switch*

*on my audio-recorder for an interview he expands: "When we have guests or family or something, I'm told to put the bloody things away and not to display them and show off, and that sort of thing". Charles grimaces as he explains that his wife, a retired GP, does not like his medicines to be visible. He goes on to say that his wife thinks they should be kept discretely out of view. As he talks me through his medicines he lifts up the little tray from the coffee table onto his lap and goes through them meticulously, one by one, though he is careful to tell me that he is approaching them in 'no particular order'. (Fieldnotes, home visit to Charles, DS)*

These quotes and ethnographic notes point not only to the ways in which space, time and physical arrangements are harnessed to routinise and support medicines taking but also how efforts at organising are a considered social performance. Our participants varied in the extent to which their medical problems were made visible through their medicines to others who might visit the home. This finding resonates with the work of Palen and Aalokke who studied medicines management among elders in assisted living apartments in Denmark.[30] A key point of difference is that in the Danish study the participants received at least daily visits from a mobile healthcare worker who collaborated with elders in their medicines management. In our study home, visits by health professionals were rare, even for our six housebound patients. Patients devised their own strategies and routines, although this often involved considerable resourcefulness and collaboration with others.

Given the number of items (10–30) and variety (tablets, creams, eye drops, injectables, inhalers and inhaled oxygen) of medicines prescribed to participants in our study their medicines regimens were inevitably very complex and needed careful organising. For example, one study participant consumed 21 tablets every morning, 15 tablets every evening and various 'as required' medicines in between; 7 of our participants were prescribed 15 or more different items of daily medication. Despite the complexity of this work, it was difficult to persuade our participants that there may be value in us learning about it as researchers. While it was clear to us that they had given much thought to how to integrate their medicines into their lives, they mostly regarded their medicines as uninteresting, routine, normal, mundane day-to-day business, but nevertheless '*a nuisance*'. Unlike their medical histories, which they often recounted in detail and in a well-rehearsed manner, they often struggled to articulate their approaches to organising and consuming their medicines ('I just take them!'). In part, this may reflect the extent to which their routines had become embodied and successfully integrated into daily life; they no longer required ongoing careful consideration.

We now present three 'telling cases',[31] vignettes distilled from detailed longitudinal narrative case studies of each study participant. We seek to strike a balance between providing sufficient rich detail to weave together our analytic insights, while being succinct enough to convey the breadth of experience and practices across different patient participants. We draw primarily on our interview

data and ethnographic field notes, with brief reference to some of our other data collection activities. Our objective in our selection of data for presentation, and in our approach to presenting these data as vignettes is to illuminate patients' hidden work of organising medicines into their lives in their efforts to adhere to complex prescribed medication regimens. These cases are not selected on the grounds of typicality but for their capacity to situate polypharmacy within a social context and challenge normative biomedical conceptualisations of polypharmacy. They speak to new ways of knowing the phenomenon of polypharmacy as it shapes the lives of patients experiencing complex multimorbidities. See box 2 (Marco); box 3 (Jackie) and box 4 (Zac).

These three cases convey a 'thick description'[32] of the nature and complexity of our study participants' medicines practices. Drawing on these cases as exemplars from our wider dataset, we now synthesise these accounts to identify some key areas of shared experience and sense making among our participants.

Jackie's account shows her days beginning and ending with a concern to monitor her medicine supplies, and the vignettes describe how these three participants have devised weekly temporal patterns of organising their DIY-dosette boxes to support daily routines. In Zac's case, trouser pockets stand in for boxes (he referred to his pockets as 'boxes' throughout one of our interviews). Zac's system allows him to adhere closely to taking his medicines at the prescribed times, which is a high priority for him. Maintaining his integrity as an active, helpful, independent, good citizen trumps any concerns he has for the integrity of the medicines in the storage conditions provided by his pockets. For Marco and Jackie, the preparation of the DIY-dosette boxes by Vicky and David, respectively, are important displays of reciprocity and part of a shared biography. In the case of Marco and Vicky, their mutual acts of organising form a tangible and important part of their ongoing support of each other.

In all cases, patients are balancing the perceived 'non-negotiable' requirement to take medicines with their desire for control over *how* they organise them into their lives. What emerges is inevitably a compromise. It is very difficult, perhaps impossible, for patients who are prescribed 10 or more separate items of medication every day to take them strictly as prescribed, even when (as in our study) they express a clear intent to do so. As previous scholars have shown, there are practices of resistance,[33] and self-regulation,[34] within an overall context of reluctant acceptance of medicines as a way of life. All of our participants regard their organisation of their medicines as an expression of their desire to retain independence as far as this is possible, and to 'manage' themselves in a context of increasing dependency. Marco wants to avoid '*bothering*' his GP; Jackie does not want to '*bother*' her neighbour about her eye drops. They all have strategies to ensure they do not run out of medicines.

Both Zac and Marco resist professional efforts to take over organising their medicines through the use of

## Box 2   Vignette of Marco

Marco is a retired chef. His narrative begins over 10 years ago at age 60 when he was diagnosed with diabetes and then with a 'cardiac problem' which led to him having major surgery. He lives in an owner-occupied flat on the top floor of a low rise block in a deprived inner city area. We usually sit outside on his covered balcony when I visit; he smokes cigarettes and sips 'real' coffee ('life's little luxuries') as he talks.

He tells me about his blackouts, painful joints, lack of youthful energy and strength, fingers which get sore from finger prick tests (which he tactfully avoids except in the week before his diabetes check-up), and his warfarin levels which 'go up and down like a bloody yo-yo'. He has no upper teeth and his dentures don't fit as he is awaiting dental treatment, so despite his fondness for cooking, eating is difficult.

Marco displays a sense of resignation about his ill health ('I don't bother the doctor with everything—what can they do?'). He is prescribed 15 items of medication but has not seen his GP for three years, although he attends the surgery regularly for nurse-led diabetes checks and warfarin monitoring blood tests ('a nuisance'). His medicines mean 'survival…I have to take them for life to keep me alive' but at the same time they 'get on my nerves' and 'I wouldn't take any of them by choice… but I don't have any choice.'

Marco's pharmacist offered to package his medicines into a MCCA and deliver them direct to his flat, but he declined:

'Sometimes you're offered a service but it doesn't mean you really need that service, you know what I mean? I can do it myself, it gives me a little exercise…it can wait a little while before I need the service…I can do it myself now.'

He continues:

'There's a couple of people in this block I noticed they have it [their medicines] delivered; they have this delivered, that delivered, when they're capable and it would be good for them to go around there and get it, you know?' He goes on: 'sometimes it's just laziness.'

I soon learn that Marco's friend Vicky helps him organise his medicines, though he tells me repeatedly that he could easily manage himself. Vicky is in her 40's and visits every day; she sometimes answers the phone when I call. She prepares his Do-It-Yourself [DIY] cassette-type MCCA on Sundays at a frequency that depends on the results of Marco's warfarin monitoring [International Normalised Ratio or INR]. If his INR is stable she may prepare four weeks at once. After all, Marco says 'you don't want to be undoing it all again do you? It would be nuisance'. Vicky also prompts him to order his medication from his GP using an online service when supplies are low. Marco says he knows 'roughly' what his tablets are, although a video-recording of one of his clinical consultations with his GP casts some doubt on how well he knows them by either name or purpose. Warfarin and insulin are the only medications he ever refers to by name—the others he knows by condition (diabetes, blood pressure), shape and size (the 'big one'), time of day (the 'night one', or organ ('heart'). This knowledge is adequate for his organisational purposes.

Marco explains that the DIY MCCA he uses currently is bigger than previous models he has tried—his medicines outgrew the smaller versions. The cassette consists of a stack of seven rectangular boxes labelled by day of the week. He takes the 'Monday' box from the bottom, causing the others to slide downwards within the supporting rack. There are 12 pills distributed across the four cells but each day he shifts the 'midday' ones that Vicky has prepared into the 'morning' cell (any time from 10 am until 1pm, it turns out) and shifts the 'teatime' ones into the 'evening' cell; two rounds of medicine-taking is easier than four. It seems that Vicky feels obliged to organise them as the prescription dictates, leaving Marco free to take responsibility for his personal

*Continued*

## Box 2   Continued

re-organisation. He tries to take his evening medicines 20 minutes before eating but if he forgets—which he admits he sometimes does—he takes them two hours after eating 'or thereabouts'. He finds it 'very, very annoying' that his medicines change in shape and colour so often, especially as he is colour blind and colours are confusing at best of times. His insulin pen sits on the coffee table, with extra supplies in the fridge. He keeps three inhalers in a 'Man Tin'. There is a glyceryl trinitrate (GTN) spray by the bed 'just in case my heart starts messing around' and he adds 'it comes out with me all the time'. In addition, he keeps a basket in the bathroom where:

'I do accumulate a little surplus, but I try not to, not so much, you know, just a little bit so it gives me an extra three or four days, besides what they give me, you know?… But I don't have a massive supply, because what happens is if she [the GP] changes one of them in strength and they won't take the tablets back or anything back… only a few, not stocking up, you know.'

He retrieves his warfarin paperwork which is tucked behind an ornament on the mantelpiece. He points to the date of his next appointment in 1 weeks' time. He explains he has to take his 'yellow book' to the surgery every time he needs a prescription for warfarin; this requirement doesn't align well with the electronic routines now in place for ordering medicines online and transferring prescriptions between his GP and pharmacist.

On a later visit, Marco shares a poignant story about Vicky. When he met her a few years ago she was unable to manage her money, was overspending and not caring for herself. Marco bought her a tin, with two compartments—one for 'this week's' money and one for 'next week's' money—a kind of 'financial compliance aid' and has helped her 'get on her feet' financially. Marco told me that he had advised Vicky that 'it is easy to live a life in which you just don't care about anything. It is harder to live a life in which you care.'

pharmacy-prepared MCCAs. They support this resistance with claims to cognitive capability. Marco's narrative points to the possibility of future decline when he may need to make use of the MCCA 'service' but for now he constructs his identity as a capable, responsible user of both GP and pharmacy services who is prudent in ensuring he accumulates 'only a little surplus' to avoid needless waste within a resource limited health system. Zac's persistent efforts to retain this position are ultimately thwarted.

## DISCUSSION

Our study illuminates the organisational work that patients with multimorbidity do as they strive to adhere to their medicines in situations of complex polypharmacy. To our knowledge, no previous study has focused in detail on the work that goes on behind closed doors in the homes of older people prescribed 10 or more items of medication. Polypharmacy requires organising, whether it occurs in the community pharmacy or at home. In both places, this is demanding, time consuming and emotionally laden work. When doctors prescribe medicines, they prescribe work, but the nature, scale and complexity of this work usually remains invisible and unknown to the prescriber. The effort required by patients to do this

**Box 3 Vignette of Jackie**

Jackie was diagnosed with a lifelong neurological condition as a teenager when she spent a year in hospital unable to walk. Her complex biography includes a stillbirth, a prolonged episode of severe loss of vision from which she recovered, numerous operations, diabetes, domestic abuse, two marriages. Now in her 70s she has ongoing chronic pain. Jackie lives in social housing with her son and is prescribed 11 items of regular medication. She begins our interview by dropping her medicines accidentally on the floor. It is not unusual for us to find pills on the carpets when we visit our research participants; they are often clearly beyond their reach. With some difficulty Jackie manages to retrieve them and explains 'I take a metformin and something that goes with the metformin beginning with an 'A'—it's on my list somewhere—they go together in the morning'. She refers to the 'thing that goes with the metformin' four times during our interview and I conclude this is how she 'knows' and remembers it, though I later learn it is also 'the orange thing'. She tells me 'I like to get them down my throat as quick as I can, because some of them have got some horrible taste to them… powdery, at the back of your throat, they lay there'. She explains that the bad taste stays for about an hour. She goes on: 'In the evening, I take a metformin after a meal, on its own, and then—about no later than 10 o'clock—I take a co-codamol and carbamazepine and something beginning with 'E' which is a cholesterol tablet. And then just before I go to bed I take my diazepam to stop me shaking in the night'. Through the day, she is also guided by her pain, adjusting her medications 'according to how the pain is' and she has other medicines which she takes to reduce the 'acid'—a symptom she attributes to 'taking so many of them [medicines]'. And then, there are the 'terrible' eye drops which she cannot manage independently because of the effects of her neurological condition on her hands:

'I can hardly get them into my eyes. It's terrible. My son has to do it, because whenever anyone comes near my eye I shut it. And I said 'how long's that for? and she [the GP] said 'for life'.

She thinks she is supposed to have her eye drops four times a day but as her son goes out to work at 06:45 there is no one to help with them during the day. A neighbour has offered but Jackie doesn't want to 'keep bothering people'.

Jackie has a faded, well-used, Do-It-Yourself MCCA—one compartment per day, the days of the week now barely visible on the lid. David, her son, fills it every Sunday. She explains that she has got it out for me to see, but usually it's kept 'out of the way' in the kitchen. Each compartment contains eight tablets (five separate items of medication). Jackie selects her tablets from each single compartment four times per day, choosing them by shape and colour. She is unsure of some of their names but, like Marco, she recognises them by their material properties. She is more than capable of filling this box herself, but David takes pride in helping with this crucial technical aspect of Jackie's care and she is grateful that 'I've got somebody keeping an eye on me'. His role in helping her with her medicines management affirms his responsibility towards her as her official carer, alongside the more personal, hands-on aspects of her care, such as turning her most nights when she calls for him as she gets stiffness and spasms in her legs.

Although it is fiddly, Jackie manages to empty her 'as required' medicines (she calls them 'her spare ones') from their blister packaging herself, and puts them together in a separate small 'spare tin': co-codamol for pain, prochlorperazine for dizziness and omeprazole for indigestion (a pill which degrades outside its original packing). Like her regular medication in her DIY MCCA the different 'spares' are mixed together. When we visit, this tin is in the kitchen, but it accompanies her to bed at night and fits neatly in her handbag for trips outdoors.

Continued

**Box 3 Continued**

In addition, Jackie has a basket under her bed where she keeps tablets in their original packs. This is where she places medicines when they are delivered to her house by the pharmacy, and she tries to anticipate her needs by staying about month ahead. She monitors her basket daily when she gets up and when she goes to bed, and calls the pharmacist on her mobile phone direct from her bed when she recognises her supplies are low. By keeping them under her bed and checking them daily she knows she won't 'get to a stage, I'd think Oh My God, I've got no tablets'. She tries to strike a compromise between running out of medication and falling foul of the rather stark warning on the repeat prescription list she has from her doctor's surgery which reads: 'Don't stock-pile medicine at home—only order what you need'. Like her MCCA, she keeps these medicines 'out of my way all day because they are upstairs'. She prefers her medicines to be out of sight when she is not dealing directly with them.

Jackie speaks fondly of her local pharmacy and says they 'know me, because they call me Jackie, so they know me really well'. But she complains that 'they [the medicines] never all run out together. It's a nightmare. I feel as though I am permanently ringing up the chemist for my tablets'. We ask Jackie if there is one thing she could change about her medicines what would this be. Without hesitation she replies 'Not take any of them!'

work, and patients' capacity to do it are rarely discussed in formal processes of medication review. This work includes: practical acts of organising; surveillance of self and supplies; creative workarounds and experimentation with prescribed regimens; evaluation and management of personal priorities; anticipation and preparedness; management of self-presentation, itself a form of 'moral' work that includes the presentation of self as a 'good' adherent patient[35]; and securing the collaboration of carers and other professionals.

Previous research has identified households as 'hybrid centres of medication practice' where medicines are assimilated and many different forms of knowledge and expertise are brought together.[36] The key focus of our study is on individual patients, rather than households, but our findings show the extent to which individual patients' practices are embedded in, and shaped by, social context and relational networks, and how patients build a particular kind of expertise in organising their medicines into their lives.

Our longitudinal study design has enabled us to locate polypharmacy within our participants' biographies and lived experiences. The 'ordinary' act of organising medicines emerges as a highly complex, deeply embodied and socially negotiated phenomenon. The patients who engage in this work typically regard it as unremarkable, mundane and taken for granted—a 'nuisance'. The ethnographic gaze unpacks the scale and scope of this 'nuisance', makes visible the tacit knowledge underpinning the work, and brings into sharp focus the connectedness of medicines to meanings, social identities, practices and relationships. These connections extend far beyond

**Box 4 Vignette of Zac**

Zac, who is almost 90 is a retired secondary school teacher. He has lots of friends, is active in his church and serves his local community by running errands and cooking for people frailer than he. He tells me he derives great satisfaction from helping others. He adopts a dutiful, independent approach to his medicines and often talks about being a 'good' patient—'my doctor tells me that I'm one of her best patients because I seem to do the right things, eat the right things and take the medication'. He tells me that people cannot believe he is the age he is: 'It is my education and my faith that have got me through…and I look after myself'.

He keeps a three-month supply of medicines in a carrier bag by his armchair in his living room. Every Saturday, without fail, he organises his medicines for the following week. He lines up his morning and evening tablets, and puts them into 14 separate small plastic bags, wrapped inside tissue paper. He then puts them into his trouser pockets: left pocket for morning tablets and right pocket for evening tablets. He keeps a whole week's supply here, taking a packet from each pocket every morning and evening. The system works well for Zac. His medicines are high priority, kept close to his person and his busy lifestyle need not interrupt his medicines taking—nor vice versa; he can keep religiously to his schedule wherever he happens to be.

Like Marco, Zac has been offered a pharmacy-prepared MCCA and has resisted ('as long as I have my faculties'). He heard on the news that people make mistakes with the doses of medication in these boxes; he doesn't want anyone 'mixing things up' and his 'brain is turning over well'.

A few months later, Zac is admitted to hospital. Some of his diabetes pills are stopped, he is advised to start insulin injections and he now receives a weekly MCCA. No one discussed the MCCA with him and he is unhappy about it, protesting 'while I am capable I want to do it myself'. He now feels 'on edge' every Wednesday and stays in his flat waiting for his medicines to drop through the letterbox. On one occasion they did not arrive until after 7 pm, after his pharmacy's advertised closing time and he became extremely worried. On another, he noticed the 'cholesterol medicine' was missing. In our cultural probe activity ('wishful thinking') he explains that he wishes he could have a conversation with his GP about going back to his old system, adding 'I want to be in control of my own destiny'. He wants to be free to go out and dislikes the weekly round of anxiety that he may run out of pills. He walked to the surgery several times hoping to secure an appointment to discuss this, only to be told there were no appointments.

Several months passed, before we observed a clinical consultation between Zac and his GP. Zac explained 'I'm not very happy with the dosette box… they restrict me in my work and my movement.' The GP explained that Zac could have more than one box delivered at a time and promptly arranged this with the practice administrator responsible for MCCAs. Zac continued cautiously: 'I'm not particularly keen on dosette boxes. I'm quite careful about medicines' but the GP did not get to the bottom of Zac's concern.

In a conversation after the appointment, Zac explained to us that pharmacists can make mistakes with the medicines 'so you still have to check that they've done it properly', suggesting that he did not feel relieved of the organisational burden but had lost the sense of control and freedom that he enjoyed before. He complained that sometimes he can't recognise the tablets now because the brands change and went on to say that people who like dosettes are 'taking the easy route'.

We realise his diligence is being undermined and he speculates 'maybe they thought now I'm over 80 I can't manage?' He tells us he will try out the 'four weekly dosettes' then go back to the GP again and ask 'why

**Box 4 Continued**

I even need to be on them [the dosettes]'. Several months later in our final discussion with Zac, before the end of our data collection period, he had yet to have this conversation with his GP.

the technical and biomedical concerns which are the typical focus of the clinic.

Our study is relatively small, involving 24 patients who use the services of three GP practices and four community pharmacies. We have prioritised depth of analysis over breadth which limits the generalisability of our findings, but has enabled a close focus on polypharmacy as 'work' that must be attended to, and offers clinicians insights into the consequences of their prescribing which are usually hidden from clinical view. Our participants were all committed to adhering as well as possible to their medicines regimens. We accept that this may not always be the case; our participants' willingness to take part in our research might be a reflection of their commitment to their medicines. However, we have shown that even this group of patients struggled to adhere to regimens exactly as prescribed and had to be very resourceful to meet the organisational demands of their treatment.

Pharmacy-prepared MCCAs or dosette boxes are often offered to patients as one way of supporting adherence in situations of polypharmacy; approximately 64 million are issued in England per year[37] although professional bodies discourage their use.[38 39] Royal Pharmaceutical Society guidance clearly states: 'The use of original packs of medicines with appropriate support is the preferred option of supplying medicines in the absence of a specific need for an MCCA as an adherence intervention'. Concerns include, but are not limited to: dispensing drugs in a MCCA is an unlicensed use of medicines; the limited evidence available (mostly from care home settings) indicates a lack of patient benefit[38]; many medicines become unstable when removed from their original packaging (and drug manufacturers are not required to test the stability of medicines repackaged in a MCCA)[38]; patients may lose confidence about their medicines; scope for waste when prescriptions change. Pharmacists have legal responsibility for assessing a patient's need for a MCCA and there have been recent calls to regulate pharmacists more heavily, to improve standards of MCCA practice and reduce the number issued.[37]

In this paper, we have focused primarily on patients who are dispensed their medicines in their original packaging, as recommended in professional guidance (although Zac, like three other participants in our study went on to receive a MCCA part-way through our follow-up). Our observations highlight some of the parallels between the work of the pharmacy in preparation of MCCAs and the work that patients must undertake when they are responsible for medicines dispensed in large numbers in original packs. All of our participants devised ways of organising medicines that they could accommodate within their

lives, often using DIY-MCCAs and often exposing their medicines to some of the same risks that underpin professional concerns about MCCAs (eg, the risk of medicines losing their integrity when removed from their packaging). Some patients valued taking ownership of organising their own medicines and expressed reluctance at handing over this responsibility to pharmacists.

Calls to improve standards of MCCA practice are welcome.[37] However, the goal of reducing their use in favour of dispensing medicines in their original packaging may fall short of the mark by failing to address the most complex issues at stake. Our work shows that the burden and risks of 'organising' polypharmacy in the service of medicines adherence always fall somewhere; the work does not go away. The most complex issue sustaining this work is polypharmacy itself. Even when robust 'evidence-based' arguments can be made for each individual item in a list of 10, 15 or more drugs, the evidence supporting such extensive polypharmacy in older people with multimorbidities is questionable.[40] Combinations of drugs prescribed according to 'single disease' guidance (and often based on trials conducted in younger populations) can be both disruptive and dangerous in this context.[6 41] Unless the professional gaze shifts intentionally towards finding ways of tackling polypharmacy itself, the 'hidden' burden of organising polypharmacy will always fall somewhere, and will always carry risk.

**Acknowledgements** We would like to thank the staff and patients who participated in this study. We would also like to thank members of our Expert Advisory Group and our patient panel for their valuable advice, and colleagues at QMUL who commented on an early draft of the manuscript.

**Contributors** DS designed the study and secured funding. Both authors jointly gathered and analysed data. DS wrote the first draft of the paper. Both authors revised and finalised the manuscript. DS is the guarantor.

**Funding** This article presents independent research funded by the National Institute for Health Research (NIHR) through a Clinician Scientist Award (DS). In addition, NF was (in part) supported by the National Institute for Health Research (NIHR) Collaboration for Leadership in Applied Health Research and Care (CLAHRC) North Thames at Bart's Health NHS Trust.

**Disclaimer** The views expressed are those of the author(s) and not necessarily those of the NHS, the NIHR or the Department of Health and Social Care.

**Competing interests** None declared.

**Patient and public involvement** Patients and/or the public were involved in the design, or conduct, or reporting, or dissemination plans of this research. Refer to the Methods section for further details.

**Patient consent for publication** Not required.

**Ethics approval** The project has ethics approval from Leeds West Research Ethics Committee (IRAS project ID: 205517; REC reference 16/YH/0462).

**Provenance and peer review** Not commissioned; externally peer reviewed.

**Data availability statement** No data are available. Our ethics approval was granted based on the anonymity of the individuals consenting to participate and our participant information sheets and consent forms reflected this. Our research participants have not consented to making our raw data publicly available.

**ORCID iDs**
Deborah Swinglehurst http://orcid.org/0000-0003-1261-9268
Nina Fudge http://orcid.org/0000-0002-7161-4355

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
