## [Reviewer comments · BMJ Open]

ARTICLE DETAILS

TITLE (PROVISIONAL)	Organising polypharmacy: unpacking medicines, unpacking meanings – an ethnographic study
AUTHORS	Swinglehurst, Deborah; Fudge, Nina

VERSION 1 – REVIEW

REVIEWER	Hughes, Carmel Queens University Belfast, School of Pharmacy
REVIEW RETURNED	31-Mar-2021

GENERAL COMMENTS	Abstract: The results section seems to focus on the patients (although I accept that ‘participants’ could be a broader category), but in the conclusion, there is reference to the work of pharmacists (e.g managing MCCAs), but they had not been mentioned in the results. It would be helpful to have more explicit reference to participants other than patients, Introduction: sets the scene well and interesting to read. A comment about the sequence of material. The authors state that they begin with a description of community pharmacy practices, so I would have expected to have seen the method pertaining to this appearing first in the Method. However, it is the patient research which is described first in the method. The authors should consider how the sequencing of material aligns across the various sections of the paper. I have further comments about the method below Method: Page 6, Line 55/55. Why do the number of interviews drop from 24 (in-depth narrative interviews) to 23 (jn-depth interviews)? Going back to my earlier comment about the community pharmacy involvement, I could not see anything specific in the method about this, unless this is covered under ‘clinical consultations (page 6, line 58/59)? There also had been reference to general practice in the Design and Setting in the Abstract (page 2, line 13/14), but there is no explicit reference to general practice in the Method. Page 7 lines 3/4 It would be helpful to know more about the cultural probes, what form they took etc. They seem a little abstract. Page 7 lines 28/29. It would be helpful for the non-specialist reader to know a little more about ‘practice theory’. And is this related to the comment in the Abstract (page 2, lines 28/20) about using a practice theory’ lens? Page 9 and 10. Initials appear after the pseudonyms used for patient names e.g DS, NF. I finally realised that these relate to the authors, and I assume refer to the person who conducted the interview. This would need to be reported in the Method to avoid confusion for the reader.
---

	I read the vignettes with great interest and they are powerful in conveying the complexity faced by patients. However, there was not always close alignment between the method and the presentation of results e.g. the 'clinical consultation', which is referred to in the method is not very prominent (or even obvious) in the results. I can see how the vignettes are trying to tell the patient's story (the case), but I am struggling to see how all of the activities outlined on pages 6 and 7 were brought together. Although there is reference made to the 'professional context', this only seems to relate to the dispensing of MCCAs, and there seems to be nothing from the perspectives of pharmacists or GPs. These may be part of another paper/publication, but I think it would be helpful to have the rationale for not including these data. However, MCCAs do seem to dominate the paper And again, in terms of aligning the method and results, on page 17 in Zac's story, reference is made to a 'cultural probe activity' (line 38). I was not clear what this was. Discussion: I felt that this was a little under-developed and there is a very strong focus on MCCAs (see earlier comment on this). There is also no recognition of the limitations of the study.
--	--

REVIEWER	Mohamed, Mostafa University of Rochester
REVIEW RETURNED	17-Apr-2021

GENERAL COMMENTS	Polypharmacy is a raising problem in older adults and adherence to these medications remains a challenging process, which could affect their health outcomes. This qualitative study explored how older patients affected by polypharmacy manage and organize their medicines, how they make sense of this work and integrate it into their lives. They used ethnographic approach patients over 18-24 months in patients' homes, general practice and community pharmacy, in England, UK. This study added to the existing literature through informing decision-making and practices around medication management, which are often hidden, from view and unknown to professionals who prescribe and dispense medicines. Findings from this study indicate areas for future research, which would be of great interest to your Journal readership. To further strengthen this paper, I have provided a few comments below for the authors to consider  1, providing some information about prevalence of polypharmacy in older adults could be helpful 2. Line 30: this is usually referred as excessive polypharmacy 3. Abstract: line 16: the authors mentioned they used mixed qualitative methods. This study looks as a complete qualitative study as did not involve a quantitative or mixed methods part. The authors should provide an explanation for using the term "mixed methods" 4. Methods: it may be helpful for the readers if the authors provide more details why they used the ethnographical methodological approach, as well as the key principles of this approach, which guided this research 5. Discussion: last paragraph: "Polypharmacy in the context of complex multimorbidity is rarely, if ever, 'evidence-based', even
--

	when a robust argument can be made for the prescription of each individual item”. This sentence is a bit weird to me. How polypharmacy is not evidence based if the individual medications are prescribed based on evidence? I suggest this sentence needs more clarification form the authors
--	---

VERSION 1 – AUTHOR RESPONSE

Reviewer 1: Prof. Carmel Hughes, Queens University Belfast

Abstract: The results section seems to focus on the patients (although I accept that ‘participants’ could be a broader category), but in the conclusion, there is reference to the work of pharmacists (e.g managing MCCAs), but they had not been mentioned in the results. It would be helpful to have more explicit reference to participants other than patients,

We thank the reviewer for the time taken to read our paper and for her valuable comments to improve our paper.

The focus of this paper is how patient participants organised their medicines. However, our wider study has given us access to the working practices of pharmacists. We want to make the point here that polypharmacy always requires considerable ‘organising’ work, whether this is done by pharmacy staff preparing dosette boxes (MCCAs) or by patients who devise their own ‘do-it-yourself’ arrangements for organising.

To make this point clearer we have added the following sentence to the abstract and clarified when we are referring to patient participants:

All patients had developed strategies and routines for organising medicines into their lives, negotiating medicine-taking to enable acceptable adherence and make their medicines manageable. Strategies adopted by patients often involved the use of ‘do-it-yourself’ dosette boxes. This required careful ‘organising’ work similar to that done by pharmacy staff as they organise medicines into multi-compartment compliance aids (MCCAs). Patient participants incorporated a range of approaches to manage supplies and flex their regimens to align with personal values and priorities

Introduction: sets the scene well and interesting to read. A comment about the sequence of material. The authors state that they begin with a description of community pharmacy practices, so I would have expected to have seen the method pertaining to this appearing first in the Method. However, it is the patient research which is described first in the method. The authors should consider how the sequencing of material aligns across the various sections of the paper. I have further comments about the method below	The point about sequencing is well made and whilst we focus primarily on patient practices in this paper we realise it is important to include a little more detail on the wider study methods in order to create a coherent overall account. Many thanks for drawing our attention to this. We have amended the paper as follows:  1) We have expanded the description of our methods in the Methods section (p6-7), in particular pointing out that our ethnographic observations include extensive periods of organisational ethnography in general practice settings and pharmacy settings. We have published a paper which illuminates our pharmacy work in much more detail and have now included this reference (this was not yet published at the time of our original submission) We hope that readers interested in learning greater detail of our methods might follow up our references to our Protocol paper (ref 22) and our paper on pharmacy practices (ref 24). 2) We have added a table (table 1) to the Methods section in which we list the different data collection methods across the wider study
Method: Page 6, Line 55/55. Why do the number of interviews drop from 24 (in-depth narrative interviews) to 23 (jn-depth interviews)?	One patient participant withdrew from the study due to a serious decline in health before we were able to conduct the second in depth interview with him. We have explained this in the new table 1: 23 in-depth interviews focused on participants' medicines and medicines practices (one participant withdrew from study due to a decline in health before this interview could be completed).
Method: Going back to my earlier comment about the community pharmacy involvement, I could not see anything specific in the method about this, unless this is covered under 'clinical consultations (page 6, line 58/59)?	The clinical consultations were encounters our patient participants had with GPs, nurses and hospital doctors (i.e. planned appointments). Our pharmacy data was organisational ethnography including shadowing staff and interviews with pharmacy staff. Hopefully Table 1 now makes this clear.

Method: There also had been reference to general practice in the Design and Setting in the Abstract (page 2, line 13/14), but there is no explicit reference to general practice in the Method.	General practices were indeed an important feature of the research design of the wider APOLLO-MM study. However, in this paper we do not draw extensively on our organisational ethnography conducted in GP settings (though do include reference to our accompanying patient to an appointment with their GP) We hope that the changes to the reporting of our Methods with the addition of Table 1 makes clear that the wider ethnographic study was designed to understand polypharmacy from the point of view of general practice, pharmacy and patients in their own homes. The key focus of this paper is on the patient experience and on their 'organising' practices. We have included some detail of our ethnographic observations in pharmacy settings regarding dosette box production, as we were struck by some of the parallels between the work of professionals and the work of patients. We have made a minor change in the Introduction to justify our rationale: In this paper, we focus primarily on the practices of older adults who live at home and are prescribed ten or more separate items of medication. To set the scene, we begin with a brief ethnographic description of practices we observed in community pharmacy as professionals engaged in the preparation of multi-compartment compliance aids (MCCAs) or 'dosette' boxes for some patients affected by polypharmacy
Method: Page 7 lines 3/4 It would be helpful to know more about the cultural probes, what form they took etc. They seem a little abstract.	See table 1, row 7, column 2 (and additional reference to Gaver paper for interested readers): Cultural probes encouraged participants to depict their lives with their medicines through a number of optional activities: (Gaver reference)

	 • giving participants a camera and asking them to take pictures of what their medicines mean to them • drawing body maps of how they are feeling and how their medicines make them feel • completing diaries of social contact • ‘wishful thinking’: participants imagined conversations they would like to have with a healthcare professional about their medicines
Method: Page 7 lines 28/29. It would be helpful for the non-specialist reader to know a little more about ‘practice theory’. And is this related to the comment in the Abstract (page 2, lines 28/20) about using a practice theory’ lens?	We have expanded our description of practice theory and hope this adds clarity (Analysis, page 9). In essence, the focus is primarily on ‘doings’ (not just taken-for-granted accounts, though with an important caveat that ‘sayings’ are a form of ‘doing’ and to practice theorists language IS a form of shared practice). ‘Embodied’ practices go beyond issues of understanding or cognition and practice theory draws attention to the interconnections between people and between people and things. We are limited by word count (there are whole texts on practice theory) and realise that this descriptor may seem quite abstract but we hope that when read in conjunction with our detailed ethnographic accounts it is possible for readers unfamiliar with this approach to appreciate its value and contribution (as a different way of seeing or appreciating social life). We now say: Our analysis is informed by practice theory^{25 26} which conceptualises practices as arrays of embodied human activity and draws attention to assemblages of people, technologies, artefacts and their interconnections in context, including shared practical understandings, ‘know-how’, skills, tacit understandings and dispositions The reviewer is correct - the practice theory lens we refer to in the abstract is the same we refer to in the methods and which we have now expanded upon.
Method: Page 9 and 10. Initials appear after the pseudonyms used for patient names e.g DS, NF. I finally realised that these relate to the authors, and I assume refer to the person who conducted the interview. This	Thank you for pointing this out – we have added a line in the methods under Setting and Data collection to clarify this:

would need to be reported in the Method to avoid confusion for the reader.	'author initials in the text refer to which researcher conducted interviews or documented field notes'
Results: I read the vignettes with great interest and they are powerful in conveying the complexity faced by patients. However, there was not always close alignment between the method and the presentation of results e.g. the 'clinical consultation', which is referred to in the method is not very prominent (or even obvious) in the results. I can see how the vignettes are trying to tell the patient's story (the case), but I am struggling to see how all of the activities outlined on pages 6 and 7 were brought together. Although there is reference made to the 'professional context', this only seems to relate to the dispensing of MCCAs, and there seems to be nothing from the perspectives of pharmacists or GPs. These may be part of another paper/publication, but I think it would be helpful to have the rationale for not including these data. However, MCCAs do seem to dominate the paper	Many thanks for this observation and we are delighted that you found the vignettes powerful conveyers of the complexity faced by patients. Ethnographic accounts are often particularly successful in illuminating complexities which are difficult to convey in numbers or 'themes' for example. We hope that our accounts convey to the reader a real sense of us 'being there' (hence inviting the reader in to the complexities) since this is a marker of quality in this kind of research. We understand your reservations and have discussed together in some detail how to address this. We have decided to leave the vignettes as they are (especially as you found them so powerful) and don't want to disturb the flow of the vignettes by introducing references to our methods section. In Table 5 (Zac), we have added a qualifier to our reference to cultural probe activity ('wishful thinking', now briefly explained in the Methods section) and have described his encounter with his GP as a clinical consultation so that this aligns more closely with the description in Methods. In addition we have taken more care on p.13 in how we have introduced the vignettes including this section: We seek to strike a balance between providing sufficient rich detail to weave together our analytic insights, whilst being succinct enough to convey the breadth of experience and practices across different patient participants. We draw primarily on our interview data and ethnographic field notes, with brief reference to some of our other data collection activities. Our objective in our selection of data for presentation, and in our approach to presenting this data as vignettes is to illuminate patients' hidden work of organising medicines into their lives in their efforts to adhere to complex prescribed medication regimens.

	We hope that this does the work of justifying this approach and making plain that these are constructions, or summaries of a large data set, drawn together to make particular analytic insights plain. There is necessarily a trade-off between depth and breadth when presenting ethnographic data of this kind. The ‘telling case’ is exactly what it says – selected to present to the reader the key insights that arise from the analysis of the data set, whilst retaining the focus on ‘the case’ and keeping the story of the participant intact. Regarding the comment about the pharmacy perspective and more details of pharmacy work, we our recently published paper on pharmacy practice is now cited/referenced.
And again, in terms of aligning the method and results, on page 17 in Zac’s story, reference is made to a ‘cultural probe activity’ (line 38). I was not clear what this was.	(see above) In Table 5 (Zac), we have added a qualifier to our reference to cultural probe activity (‘wishful thinking’, now briefly explained in the Methods section) We hope that the changes we made in the methods section with the addition of table 1 have clarified what the cultural probe activity was in sufficient detail.
Discussion: I felt that this was a little under-developed and there is a very strong focus on MCCAs (see earlier comment on this). There is also no recognition of the limitations of the study.	Many thanks for encouraging us to develop this further. We have expanded our Discussion (to the extent possible given word limits). In particular we include a new section on limitations of the research (and have added a bullet point in the paper Highlights reflecting this) and we have clarified the statement that Reviewer 2 (see below) found confusing. We have retained the information about MCCAs as we are making the point that whilst professional/policy guidance in this area tends to focus on reducing the use of MCCAs, simply focusing on reducing MCCA use in favour of dispensing medicines in original packs seems to slightly miss the

	point. If patients are prescribed 10 or 15 separate items of medication it is inevitable that they will need to devise ways of organising them, which brings with it its own risks (and continues to expose medicines to the integrity issues that arise when drugs are removed from packages) – it is the polypharmacy that is at the crux of the problem, and we hope that we have done a decent job of drawing clinicians’ attention to the burden of organising that polypharmacy entails for patients, encouraging them to think differently about prescriptions as prescription of ‘work’ and encouraging them to think carefully before adding further to the complexity of already complex organising work.
Reviewer 2: Dr. Mostafa Mohamed, University of Rochester	
Polypharmacy is a raising problem in older adults and adherence to these medications remains a challenging process, which could affect their health outcomes. This qualitative study explored how older patients affected by polypharmacy manage and organize their medicines, how they make sense of this work and integrate it into their lives. They used ethnographic approach patients over 18-24 months in patients’ homes, general practice and community pharmacy, in England, UK. This study added to the existing literature through informing decision-making and practices around medication management, which are often hidden, from view and unknown to professionals who prescribe and dispense medicines. Findings from this study indicate areas for future research, which would be of great interest to your Journal readership. To further strengthen this paper, I have provided a few comments below for the authors to consider	We are very grateful to the reviewer for reading our paper and for her positive evaluation.
1, providing some information about prevalence of polypharmacy in older adults could be helpful	We have added brief reference to the increasing prevalence of polypharmacy, which we hope conveys the sense of it being of pressing importance. Word count precludes more detailed description of the phenomenon although we have published this before in a number of editorials and in our protocol paper (all referenced in the bibliography if readers are interested)

2. Line 30: this is usually referred as excessive polypharmacy	We realise there are interchangeable terms to describe phenomena such as the excessive/inappropriate/high risk prescription of medication. We are using terms from a King's Fund report which differentiates between appropriate and high-risk polypharmacy. Therefore we prefer to keep the wording as is.
3. Abstract: line 16: the authors mentioned they used mixed qualitative methods. This study looks as a complete qualitative study as did not involve a quantitative or mixed methods part. The authors should provide an explanation for using the term "mixed methods"	We agree with the reviewer that using the term 'mixed methods' is confusing as generally that refers to study design using quantitative and qualitative methods. We have therefore removed the term from the abstract to avoid confusion.
4. Methods: it may be helpful for the readers if the authors provide more details why they used the ethnographical methodological approach, as well as the key principles of this approach, which guided this research	In our introduction (p.5) we include a short section explaining the value and some key principles of ethnographic approaches and include two relevant references. Ethnography is well suited to investigating complex multifaceted phenomena, making tacit knowledge and practices visible.²⁰ It involves immersion in the field, an attitude of 'appreciation' towards the social world as it is, presentation of 'first hand' accounts of observations, and acceptance that research is an active process in which accounts of the world are constructed through selective observation and theoretical interpretation
5. Discussion: last paragraph: "Polypharmacy in the context of complex multimorbidity is rarely, if ever, 'evidence-based', even when a robust argument can be made for the prescription of each individual item". This sentence is a bit weird to me. How polypharmacy is not evidence based if the individual medications are prescribed based on evidence? I suggest this sentence needs more clarification from the authors	Thank you for pointing out that this sentence isn't clear. We were referring to the fact that the way drugs are trialled and tested is usually on younger adults who do not have multiple health conditions. Therefore, whilst each drug may be evidence base for use on its own in the context of the 'single disease model', this foundation of evidence becomes questionable in the context of older people with complex collections of conditions who are consuming multiple medications. We have reworked this statement and expanded slightly as follows (p.23): The most complex issue sustaining this work is polypharmacy itself. Even when robust 'evidence-based' arguments can be made for each individual item of medication in a list of ten, fifteen or more medications, the evidence supporting such extensive

	polypharmacy in older people with multimorbidities is questionable. ⁴⁰ Combinations of drugs prescribed according to 'single disease' guidance (and often based on trials conducted in younger populations) can be both disruptive and dangerous in this context. ^{6 41}
--	--

VERSION 2 – REVIEW

REVIEWER	Hughes, Carmel Queens University Belfast, School of Pharmacy
REVIEW RETURNED	15-Jul-2021

GENERAL COMMENTS	Many thanks for the detailed and thoughtful consideration of my comments, your response and revised manuscript. I have no further comments on this paper.
---

REVIEWER	Mohamed, Mostafa University of Rochester
REVIEW RETURNED	04-Aug-2021

GENERAL COMMENTS	Thank you for your comprehensive replies and considering the comments/ concerns which made the manuscript in a better shape and improved its quality.
---